# In Situ Characterization of Municipal Solid Waste Using Membrane Interface Probe (MIP) and Hydraulic Profiling Tool (HPT) in an Active and Closed Landfill

M. Sina Mousavi, Yuan Feng, Josh McCann and Jongwan Eun *

Civil and Environmental Engineering, University of Nebraska-Lincoln, Lincoln, NE 68588, USA;
sina.mousavi@huskers.unl.edu (M.S.M.); yuan91@huskers.unl.edu (Y.F.); jmccann924@huskers.unl.edu (J.M.)
* Correspondence: jeun2@unl.edu; Tel.: +1-(402)-554-3544

**Abstract:** Municipal solid waste (MSW) landfills near a metropolitan area are renewable energy resources to produce heat and methane that can generate electricity. However, it is difficult to use those sources productively because disposed MSW in landfills are spatially and temporally heterogeneous. Regarding the prediction of the sources, the analysis of in situ MSW properties is an alternative way to reduce the uncertainty and to understand complex processes undergoing in the landfill effectively. A hydraulic profiling tool (HPT) and membrane interface probe (MIP) test measures the continuous profile of MSW properties with depth, including hydraulic pressure, temperature, electrical conductivity (EC), and the relative concentration of methane at the field. In this study, we conducted a series of the tests to investigate the MSW characteristics of active and closed landfills. MIP results showed that the methane existed closer to right below the top cover in the active landfill and several peak concentrations at different layers of the closed landfill. As the depth and age of the waste increased, the hydraulic pressure increased for both landfills. The average EC results showed that the electrical conductivity decreased with the landfill age. The results of hydraulic properties, temperature, and EC obtained from active and closed sites could be used to estimate the waste age and help designing energy recovery systems.

**Keywords:** municipal solid waste; landfill; temperature; renewable energy; material characterization; sustainability; environmental impact



## 1. Introduction

With population growth and urbanization, municipal solid waste (MSW) generation continues to increase, which has resulted in socio-economic and environmental problems; therefore, the efficient management and possible energy of MSW landfill recovery have become focuses of infrastructure design [1]. Modern MSW landfills, in particular near a metropolitan area, are being viewed as significant sources of renewable energy to generate electricity by taking advantage of a great amount of heat and biogas (e.g., methane) that are generated from the biological activity in landfills. The ongoing biodegradation process of organic material in landfills during their lifetimes generates methane and heat.

Modern regulated landfills in the U.S. collect around 2.6 million tons of methane gas that produces thermal and electrical energy that is equivalent to a capacity of 50 MW of turbine energy. There are different technologies to convert gas to electricity including gas engines, gas turbines in the Brayton cycle, the Organic Rankine cycle, and Stirling cycle engines [2–4]. However, trace materials (including hydrocarbons, siloxanes, and volatile halogenated hydrocarbons) in landfill gases are reported to impact engine performance [5]. Landfills have also been identified as a source for geothermal heat energy. Studies have shown that temperature elevated to more than 60 °C in most typical MSW landfills, and there are also reports of landfills that pass a temperature of 120 °C [6,7]. Coccia et al. [8] demonstrated that adding a plumbing system for heat exchange would not result in a high

cost for landfill managements. Their research indicated that this system is a sustainable method to be applied in a practical way [8]. Several case studies have shown the efficiency and cost-effectiveness of landfill-based geothermal systems that can collect enough heat from waste compared to those of conventional ground-based geothermal heating systems [9].

It is necessary to characterize the MSW mechanical, chemical, biological, thermal, and hydraulic properties in a landfill to be able to more accurately predict the energy resource potential and find the optimal usage. There have been many efforts to characterize MSW properties in laboratories [10,11]. However, the samples are mostly disturbed, and the methods to obtain the properties are time-consuming and costly. It is very difficult to productively use estimated MSW landfill energy sources because active and post-closure MSW landfills are spatially and temporally heterogeneous and are exposed to complex environments with varying pressures and moistures. For instance, various chemo-physical properties such as moisture content, temperature, hydraulic conductivity, electric conductivity, and concentration vary with the depth of a landfill, as well as time, which results in changing landfill gas emission rates and changing thermal potentials of MSW at different times and places. Thus, the analysis of MSW properties via an in situ testing methods is an effective way to reduce uncertainty and to better understand complex processes undergoing in a landfill. However, there have been very few efforts to conduct in situ tests to characterize disposed waste in a landfill. Yuen et al. investigated the moisture content of MSW landfills by using a neutron probe in a macroscopic scale [12]. Li and Zeiss used time domain reflectometry (TDR) probes to measure the moisture content in a landfill. For this purpose, they calibrated the sensors with MSW and reduced the effect of varying leachate electrical conductivity [13]. Recently, Plocoste et al. [14] used a portable time-of-flight mass spectrometer to measure six different volatile organic compounds (VOCs) in various stages of waste degradation. Hartwell et al. [15] conducted an in situ large diameter borehole assessment in an MSW landfill in Nebraska to characterize physical and biological properties of the landfill with depth; however, the methodology can only be used during the installation of gas collection systems. Overall, the existing experimental efforts in landfills have lacked feasibility, cost-effectiveness, and convenience, and most cases have not provided reliable data to analyze various MSW properties with depth.

The purpose of this study was to comparatively analyze MSW properties with depth using a feasible and convenient in situ testing instrument called a membrane interface probe (MIP) and hydraulic profiling tool (HPT) in active and closed landfills. Through a series of tests using a probe pushed to great depths in the landfills, hydraulic pressure, temperature, electrical conductivity (EC), and the relative concentration of methane profile were comparably measured to assess anaerobic activity and analyze the hydraulic property of the waste column.

## 2. Methods

### 2.1. In Situ Testing Apparatus

A direct injection logger, which included an HPT and an MIP enabled the measurement of different in situ parameters of MSW such as hydraulic pressure, electrical conductivity, hydraulic conductivity, temperature, and the relative concentration of selected VOCs and methane in the field. Figure 1 shows the HPT and the MIP (38 mm diameter) that operated along a drilling rig (7822 DT combo rig, Geoprobe Inc., Salina, KS, USA).

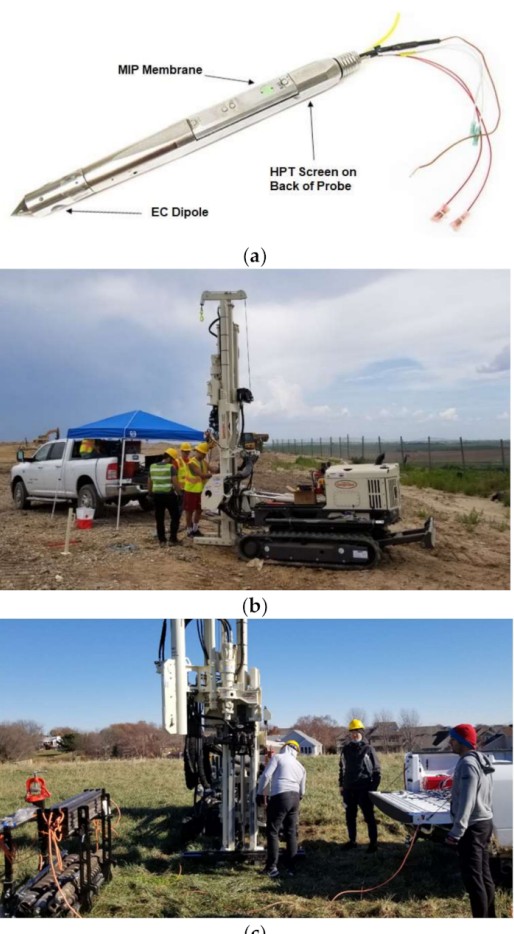

**Figure 1.** (**a**) Hydraulic profiling tool (HPT) and membrane interface probe (MIP) probe (Geoprobe.com); (**b**) drilling rig at the Butler County landfill; (**c**) drilling at the Sarpy County landfill. EC: electrical conductivity.

An HPT with an MIP (known as an MiHPT) consists of a direct injecting cone-device, protective sleeve tubes attached with sensors, and a large drilling machine. The HPT sprays fluid at a high pressure to the adjacent material as the probe is driven into the subsurface and determines the hydraulic conductivity by detecting the release of hydraulic pressure within the material. An MIP defines the continuous profiles of selected landfill gases (LFGs), mainly methane in a landfill. An MIP consists of two main parts: the surface instrumentation and downhole probes, as shown in Figure 2. An MIP can collect gaseous samples from a subsurface and transfer the sampled vapor to the detectors equipped on a gas chromatograph (GC) on the surface and give a real-time distribution profile of VOCs with increasing depth. There is a permeable membrane that acts as the interface to the detector line, and a carrier gas (e.g., nitrogen) transfers the contaminant to the GC detector. Two detectors—a photoionization detector (PID) and a flame ionization detector (FID)—are installed in a GC and are used to detect the presence of organic compounds. Each detector is sensitive to a specific group of compounds. For example, an FID detects hydrocarbons, and an increase in an FID with a small decrease in a PID indicates the presence of methane. Before the test is started, a pre-log test is conducted with the expected VOCs (e.g., pure methane) to calibrate and assure the detecting responses to the compounds. A heater panel is used to increase temperature up to 120 °C to help with a gas extraction. An MIP can work in all types of materials and can be applied in unsaturated conditions [16].

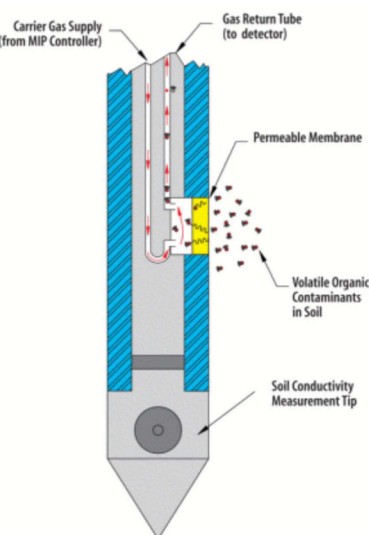

**Figure 2.** A schematic view of an MIP modified from [17].

An HPT has an up to 700 kPa pressure transducer, which is equivalent to approximately 70 m water head. An HPT logs one piece of data for every 15 mm for HPT pressure, flow, and electrical conductivity. An HPT probe also measures hydrostatic pressure under the zero-flow condition. The piezometric profile is then used to correct the obtained HPT pressures [16].

### 2.2. Testing Site and Procedure

In this study, two sets of the MiHPT tests were conducted in two landfills. The first test was performed at an active landfill, and the second was at a closed landfill. The first set of tests (including two tests at one site) was conducted at the Butler County landfill, Nebraska, U.S.A. The landfill, which is located close to the Omaha metropolitan area, accepts municipal solid waste, industrial waste, construction and demolition waste, scrap tires, special waste including asbestos, and non-hazardous contaminated soils. The total depth of the cell was 40 m, and it had been active since June 2008. The top interim cover for the site was a gently compacted silty clay soil with a low plasticity. During the test, weather conditions were 32 °C with no rain in the past 3 days. Two holes were drilled approximately 4.0 m from each other. The probes were advanced to depths of 14.6 and 17 m. The membrane was left working for around 45 s at each depth interval (approximately 0.5 m), as per the recommendation from the user manual. In the first tests at the Butler County landfill, both the MIP and HPT sensors were active, as was a heater panel. In the second test, the temperature profile along the depth was obtained while the heater panel on the MIP and water pump on the HPT were turned off. The temperature was obtained from the thermal gauge attached to the probe. A profile of EC was also obtained from both tests.

The second sets of tests were conducted at the Sarpy County landfill, Nebraska, USA. The Sarpy Count landfill is located in the city of Bellevue in southern Omaha but has been operated by Sarpy County since 2002. The landfill stopped receiving waste in 1999 and permanently closed in 2002. (discussed with the city environmental engineer). The final cover was a 0.6 m-compacted clay layer of 70–80% density. Most of the MSW came from surrounding residences, so the constituent of the waste was similar to the Butler County landfill. At each site, two tests were conducted 1.5 m apart. During the tests, the temperature was around –7 °C with no rain in the past 3 days. The probes were advanced to depth of around 13 m in all tests. The testing procedure of the second set of tests, including probe advancement speed and time intervals, was similar to those of the first sets of tests. The heat panel was turned off once at each site to measure the temperature profile. The EC profile was obtained three times at each site.

## 3. Results and Discussion

### 3.1. VOC Results

Figure 3 shows the FID and PID profiles obtained from the active landfill of Butler County. The MIP results at the Butler County landfill showed a spike in the FID and simultaneous changes in the PID (a decline in FiD and relatively constant measurements in PID), which indicated the presence of methane at the top layer of the landfill. Methane has a lower molar mass of 16 mol/g compared to the 29 mol/g of dry air and therefore the gas moved upward and became trapped beneath the top cover soil cap. However, the lower depth of the landfill (below around four meters) showed a very low VOC concentration. This was possibly because of the saturated conditions that emerged from the leachate recirculation practice in the landfill. It can be seen from Figure 3 that the high temperature of the heater panel assisted with the collection of VOCs other than methane.

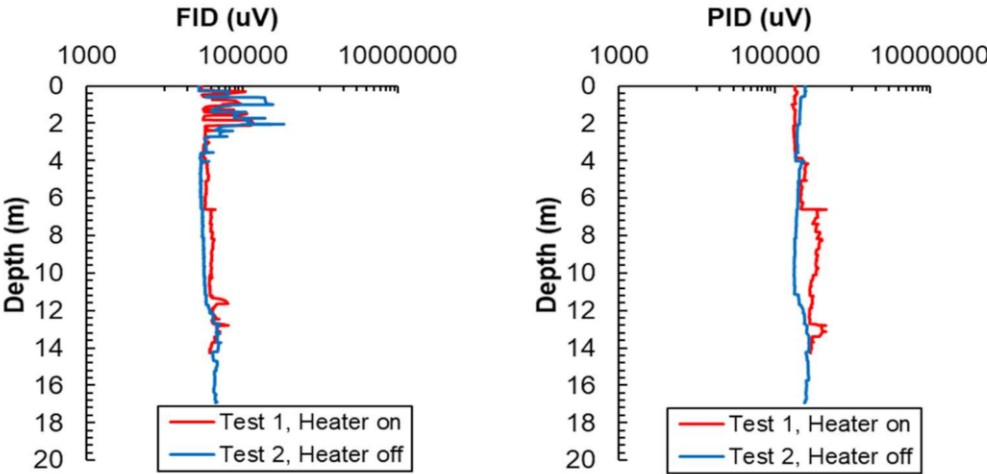

**Figure 3.** Flame ionization detector (FID) and photoionization detector (PID) responses for the MIP test at the Butler County landfill.

Figure 4 shows the FID and PID responses at the closed Sarpy County landfill. Methane peaks can be seen at the depths of around 6.0 and 11.0 m, respectively, in the first test at site 1 (Figure 4a), where the FID response showed an increasing peak and the PID response showed a small decline. It can be understood that the stratification nature of the MSW landfill and the presence of intermediate compacted soil layers resulted in methane gas being trapped beneath layers. In comparison to the Butler County landfill, the closed Sarpy County landfill had gone through years of the biodegradation settlement of placed MSW and the consolidation of soil layers, so the consolidated layers with lower diffusivity prevented the migration of methane to the top layer. The second test at the same location with the heater panel on (Figure 4b) showed the presence of a variety of VOCs, including methane, since FIDs respond to most VOCs given high enough concentration. Previous studies have shown that multiple fractions in MSW produce VOCs during the biodegradation process [18,19]. PID responses are usually indicators of the detection of polycyclic aromatic hydrocarbons (PAHs), but PAH is a product of MSW pyrolysis, which is the chemical decomposition of solid material in high-temperature landfills [20,21]. The higher responses of the PID and the FID when the heater panel was on (Figure 4) were indicators of the presence of VOCs other than PAH, which could have included malodorous sulfur compounds, aromatics, and chlorines at the Sarpy County landfill.

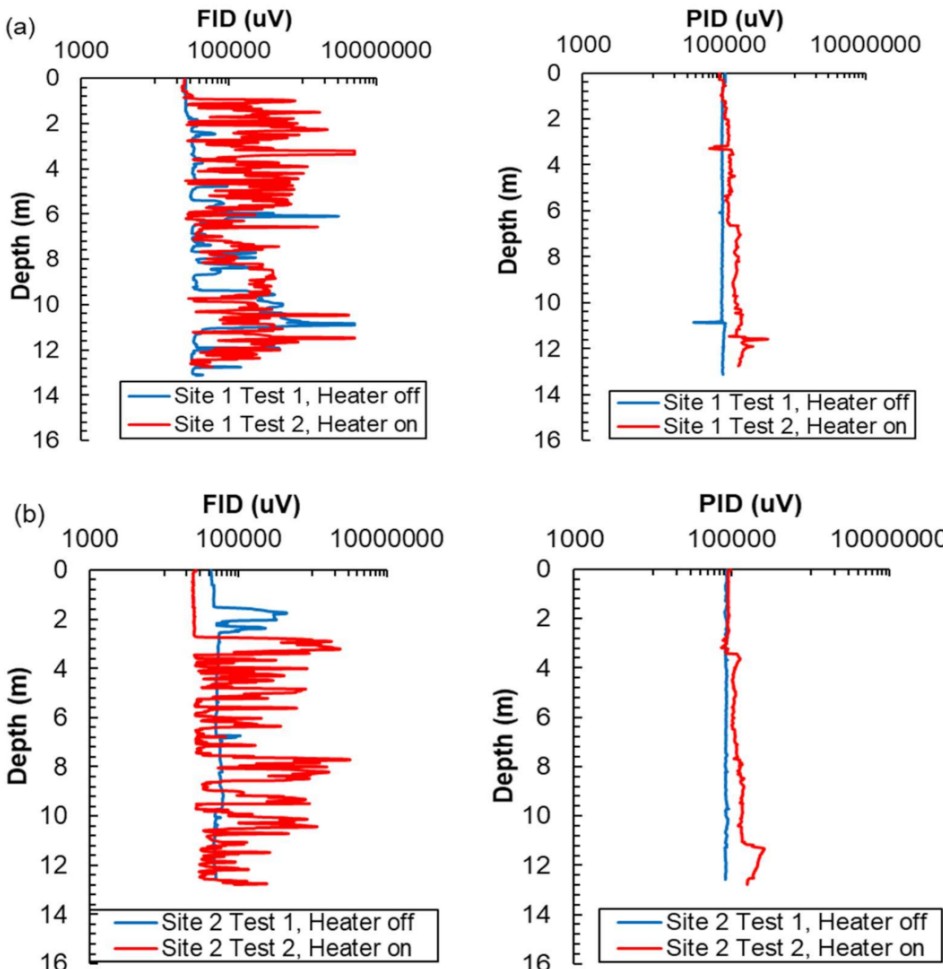

**Figure 4.** FID and PID responses for the MIP test at Sarpy County landfills: (**a**) Site 1 and (**b**) site 2.

Compared to those of Sarpy County, the results showed that the Butler County landfill, as relatively fresh and active landfill, had either not produced enough VOCs at the time of the in situ test or did not have VOCs that had been collected along with the leachate through the leachate recirculation process. On the other hand, the Sarpy County landfill, as an old landfill with no leachate recirculation system, showed the presence of VOCs compounds within the waste profile. The results from the second test at site 2 showed similar indications. Figure 4 shows the FID and PID responses while the heater panel was off. It can be seen that methane peaks emerged at the depths of around 2.0 and 6.7 m, respectively. On the other hand, when the heater panel was on in the second test at site 2 (Figure 4b), the responses of the FID and the PID showed the presence of VOCs below the depth of 2 m. The results from the active and closed landfills showed that the MIP could be a useful tool to estimate the degradation process of a landfill. In the case of the Sarpy County landfill, the presence of methane indicated that the biodegradation process of the landfill was still active, albeit past its peak. Therefore, there was more settlement induced by biodegradation activity in both sites.

### 3.2. Temperature Profile

Heat is generated from the aerobic and anaerobic conversion of cellulose $(C_6H_{10}O_5)n$ in the organic fraction of buried waste to biogas according to the following reactions [22]:

$$\text{Aerobic: } (C_6H_{10}O_5)n + (6nO_2) \rightarrow (6nCO_2) + (5nH_2O) \tag{1}$$

$$\Delta H = -17360 \ kJ/(kg \ cellulose)$$

$$\text{Anaerobic: } (C_6H_{10}O_5)n + (nH_2O) \rightarrow (3nCH_4) + (3nCO_2) \tag{2}$$

$$\Delta H = -1673 \; kJ/(kg \; cellulose)$$

Aerobic and anaerobic biodegradation processes inside landfills release heat to the surroundings, and due to the low thermal conductivity in MSW waste, heat accumulates and temperatures rise mostly in mid-depth of landfills.

Temperature profiles along the waste depth were measured when the heater panel was turned off on the MIP. As shown in Figure 5 at the Butler County landfill, the temperature appeared to decrease to the depth of 6 m, which could have been due to the accumulation of hot gases on the top surface coupled with the raised temperature of the top cap from sun exposure. However, the temperature increased up to 60 °C as the depth increased from 11 to 17 m, which was consistent with previously reported landfill temperature measurements [23,24]. The trend of the increasing temperature seemed maintained at deeper depths.

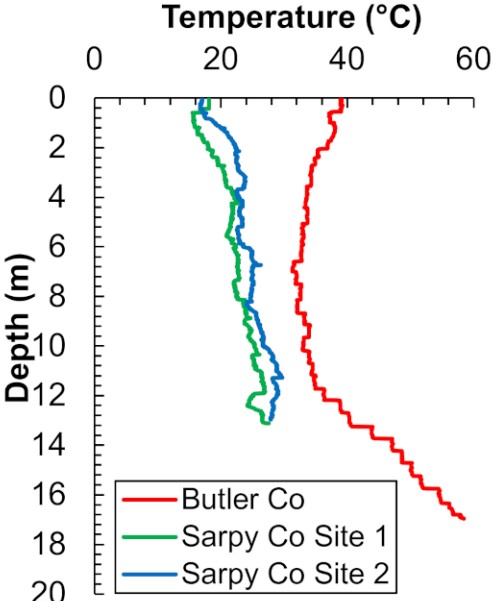

**Figure 5.** Temperature changes with depth.

The temperatures measured in both sites of the Sarpy County landfill showed similar trends. The thermocouple on the MIP was at a temperature of less than 20 °C before starting the measurement. The results showed that the temperatures slightly increased up to around 27 and 29 °C at sites 1 and 2, respectively. The lower temperature at Sarpy County was expected since the closed landfill was past the active biodegradation phase of MSW; however, a higher than ambient temperature indicated that the biodegradation process was not completed within the landfill, which confirmed the finding of the MIP profiles even a few decades since closing the landfill. It must be noted that the maximum recorded temperature at Sarpy County was beyond the mid-depth thickness of the landfill, and the overall thickness was lower than the Butler County landfill site.

### 3.3. Hydraulic Pressure Profile

The results of the HPT measurement are shown in Figure 6 for the Butler and Sarpy County landfills. Figure 5 shows that the pressures generated by injecting water at a relatively constant rate into the material by pumping water in the HPT flow module changed with the penetrating depth during the HPT test, where the downhole pressure sensor measured the pressure. The data were logged every 15 mm and continuously displayed as the probe was pushed down.

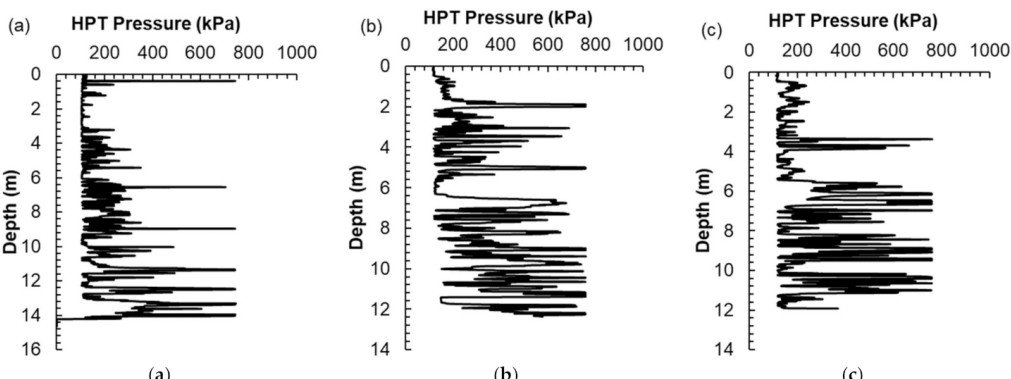

**Figure 6.** Profile of HPT pressure: (**a**) Butler County, (**b**) Sarpy County site 1, and (**c**) Sarpy County site 2.

The HPT profile in Figure 6a of the Butler County landfill site showed an increase in the HPT pressure with an increase in depth. HPT pressures were less than 200 kPa below 4.0 m and less than 350 kPa at 8.0 m. However, depths lower than 8 m showed a significant increase in pressure, reaching over 700 kPa. Large peaks separated by lower pressures indicated the heterogeneity of the waste material, where low and high permeability materials existed within the profile. The lower permeability of the materials in the MSW resulted in higher HPT pressures; therefore, it can be seen that at depths below 6−7 m, active biodegradation and higher vertical pressure resulted in the densification and breakdown of materials respectively, and, thus, a smaller void ratio that consequently resulted in a higher HPT pressure. The temperature profile of the same location showed a higher temperature as an indication of its higher biodegradation activity, which was consistent with the higher HPT pressures caused by biodegradation.

Similar trends were obtained at Sarpy County, as shown in Figure 6b,c. The HPT pressure average increased with depth; however, the overall HPT pressures were higher at Sarpy County than Butler County. Figure 7 shows the average HPT pressure versus depth for the three sites. It can be seen that the average HPT pressure increased with a linear trend. The HPT pressure was higher at all depths at the Sarpy County sites compared to the active landfill of Butler County. This was attributed to the fact that higher zones of MSW at Butler County were fresh waste that had yet to undergo biodegradation and therefore had a higher permeability. However, the Sarpy County landfill, as a closed landfill, showed lower permeabilities because of the longer time for biodegradation.

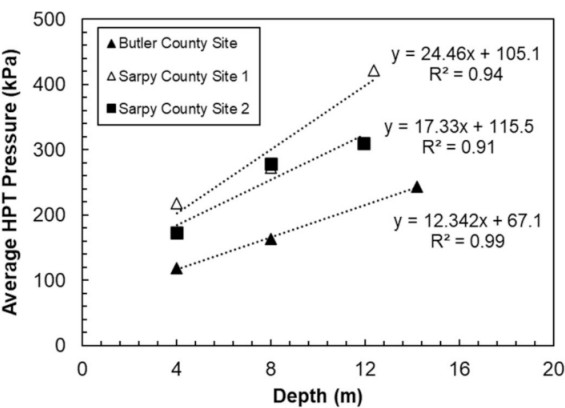

**Figure 7.** Average HPT pressures with depth.

Previous studies have conducted laboratory and field hydraulic conductivity tests for MSW and have shown that MSW hydraulic conductivity decreases with biodegradation and compaction [25–27]. However, the reported values showed that lower levels of the

landfill, where material had become denser, had lower hydraulic conductivities. Landva and Clark (1986) reported hydraulic conductivities ranging from 4.32 to 22.46 m/day in four Canadian landfills [28]. Jain et al. conducted a borehole permeameter down to 18 m below the surface and found the hydraulic conductivity to range between 0.004 and 0.05 m/day [29]. The reported values showed that there is a variety of ranges for hydraulic conductivities, thus indicating that the hydraulic conductivity of MSW is site-specific.

During the HPT tests, two dissipation tests were performed to obtain hydrostatic pressures. At these points, the water pump was turned off, and the absolute static pressure graph was obtained. An empirical equation to estimate the hydraulic conductivity is proposed in Equation (4) by [30] based on field tests and corrected pressures. The corrected pressure was defined considering hydrostatic pressure as:

$$P^* = P_{max} - P_{atm} - P_{hydro} \tag{3}$$

$$K_{est} = 6.44 \times \ln(6.896 \times Q/P^*) - 12.71 \tag{4}$$

where $P_{max}$ is the hydraulic pressure, $p_{atm}$ is the atmospheric pressure measured during the pre-test, $P_{hydro}$ is the hydrostatic pressure obtained through the dissipation test all in kPa, and $K_{est}$ is the hydraulic conductivity in m/day estimated by the empirical equation. Hydraulic conductivities versus $Q/P^*$ were plotted, where Q is HPT flow in mL/min. Figure 8 shows the results of the estimated hydraulic conductivity with increasing depth at the three sites.

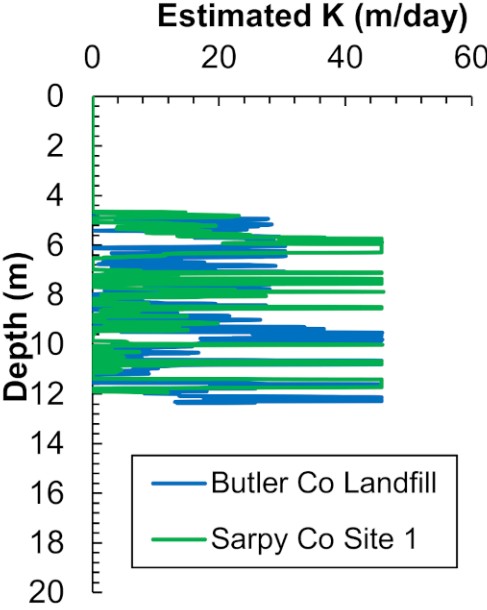

**Figure 8.** Estimated hydraulic conductivity changes with depth.

The results in the Butler County landfill (Figure 8) showed that the hydraulic conductivity had an average of 15.12 m/day between 4.9 and 8 m of depth, and the depths between 8 and 14 m showed an average $K_{est}$ of 11.98 m/day. A similar trend in average hydraulic conductivity change was observed at Sarpy County, where an average of $K_{est}$ of 16.75 m/day at site 1 was decreased to 8.4 m/day. The decrease of the average hydraulic conductivity was consistent with the lower permeability finding from the hydraulic pressure profile.

*3.4. Electrical Conductivity Profile*

An EC profile was obtained at a vertical resolution of 0.02 m [17]. Figure 9 shows the result for electrical conductivity at the Butler County landfill and two sites at the Sarpy County landfill. The results showed that the EC values were generally very low due to

the absence of materials with high electrical conductivities. However, there were high spikes in the raw data profile obtained at the landfill sites, which indicated the contact of metal debris, slag, or highly ionic materials with the EC sensor. The obtained EC values were significantly lower than the reported EC values of the MSW material, possibly due to the presence of leachate or a small surface area contact. Leachate has a low electrical conductivity, below 150 mS/m at room temperature, which generally rises to around 2% for every 1 °C increase of temperature [31].

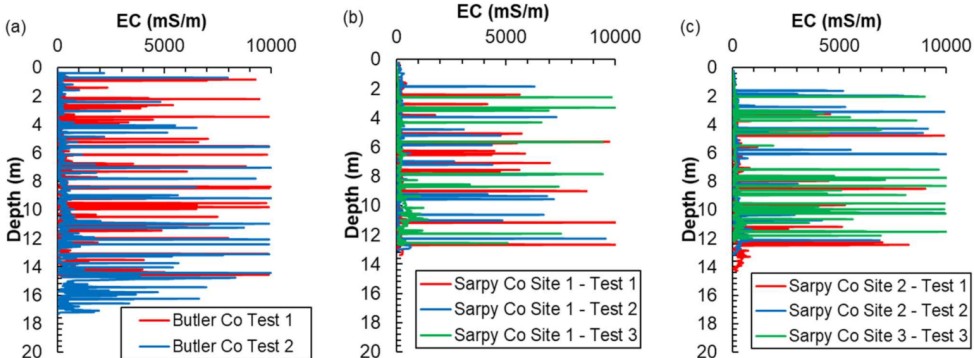

**Figure 9.** Electrical conductivity profile: (**a**) Butler County, (**b**) Sarpy County Site 1, and (**c**) Sarpy County site 2.

Upon further analysis, the high spikes from the possible contact with metals at the landfill site were considered to be outlier data. Outliers were calculated using the box plot graph method and were found to be values of approximately >800 mS/m for the Butler County landfill; 402.3, 416.7, and 457 for Sarpy County site 1; and 544.9, 519.4, and 329.3 for Sarpy County site 2. Outliers were excluded from the profile data, and 4 m intervals were selected to find the integrated average values of EC, as shown in Figure 10. The results from both Figures 9 and 10 show that the average EC values were lower in the closed Sarpy County landfill than in the active landfill. It can be concluded that lower temperatures and higher decomposition of organic material at the Sarpy County landfill resulted in a lower electrical conductivity of its MSW [32].

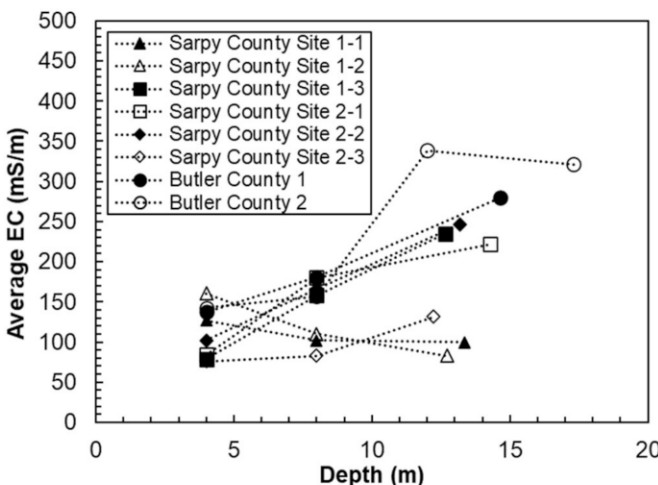

**Figure 10.** Average electrical conductivity by depth.

The trend of the average EC showed that the waste EC increased with depth in most cases. The EC results were consistent with leachates' low electrical conductivity of 150 mS/m reported by [31]. The increase in EC value could have been related to both the increase in temperature and the higher density of the material, which resulted in a higher

surface contact with the EC sensor. The results confirmed the reliability of the temperature and HPT pressure sensor, as it showed higher pressures because of densified material at lower depths.

## 4. Conclusions

Direct penetration tests were conducted at an active (with leachate recirculation) and closed MSW landfill site at the Butler County and Sarpy County landfills, respectively, in Nebraska using an MIP and HPT apparatus to develop a vertical profile for MSW temperature, VOC relative concentration, electrical conductivity, hydraulic pressure, and estimated hydraulic conductivity. The results of the MIP test showed the presence of peak methane gas concentration only beneath the top cover layer of the active landfill, while the methane peak concentration in the closed landfill was found at multiple layers at mid depth. The tests also showed a higher amount of VOCs (non-methane) at the closed landfill, which indicated a greater decomposition process over years. The MIP also measured the temperature while being advanced through MSW layers, and the results showed that the active landfill had an increasing temperature trend starting from around ambient temperature to around 60 °C at 16 m. The results at the closed landfill also showed a gradual increasing in temperature, but only up to around 28 °C, which indicated that the landfill had not reached the complete biodegradation of organic material. In addition, the hydraulic pressure and electrical conductivity of MSW was measured with depth. The hydraulic pressure and, subsequently, hydraulic conductivity increased at lower depths, which indicated the presence of densified materials at lower depths due to higher vertical stress and material breakdown from biodegradation. The average HPT pressures of the active landfill and two sites of the closed landfill increased from 118, 217, and 173 kPa at the upper layers to around 243, 421, and 310 kPa at the lower layers, respectively. The pressures were higher at the closed landfill, which further confirmed the hypothesis. The results of the EC measurements also showed meaningful changes with depth and waste age. Lower depths at Butler County and Sarpy County showed generally higher average EC values at lower depths, which coincided with higher temperature and surface area contact with the EC sensor. However, the older age of the waste clearly showed a lower EC, which was related to the decomposition of organic acids.

The results of these temperature, methane concentration, and hydraulic profiles could be useful for improving the design of gas collection systems, leachate collection and recirculation systems, and potential thermal and electrical energy conversion systems. Furthermore, MSW landfills have reported substantial uneven settlement and odor problems years after closing. The estimation of the completion of MSW landfill activity is useful information for urban planning, infrastructure management, and waste recovery designs. This study had shown that the use of an MIP and HPT can be a cost-effective, convenient, and reliable in situ testing apparatus for profile variable changes in a heterogeneous field site to evaluate the activity of an MSW landfill.

**Author Contributions:** Field testing, data analysis, and writing: M.S.M.; field testing and data analysis and curation: Y.F.; conceptualization, writing, editing, and supervision: J.E.; field testing and data acquisition: J.M. All authors have read and agreed to the published version of the manuscript.

**Funding:** This research was funded by Nebraska Environmental Trust (grant number 18-166) and Collaboration Initiative Seed Grant (15838) of University Nebraska Board of Regents.

**Informed Consent Statement:** Not applicable.

**Data Availability Statement:** The data presented in this study are available on request from the corresponding author.

**Acknowledgments:** The authors are thankful for the allowance of the site access and field tests from Sarpy County and Butler Co. landfill.

**Conflicts of Interest:** The authors declare no conflict of interest.

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
