# Peer review of "In Situ Characterization of Municipal Solid Waste Using Membrane Interface Probe (MIP) and Hydraulic Profiling Tool (HPT) in an Active and Closed Landfill"

_infrastructures, doi:10.3390/infrastructures6030033_

Round 1

Reviewer 1 Report

The proposed manuscript is interesting and useful for energy recovery from organic waste landfills.

I would suggest eliminating formulas 1) and 2) from the Introduction by finding a better location.

Line 52: managers?

Author Response

We are thankful to the reviewer for taking the time and effort in reviewing the manuscript and their comment. The responses were prepared in the attached file. 

Reviewer 2 Report

This study deserves recognition for the detailed data obtained on landfill sites.

However, I would like you to consider amending or adding the following points.

L46-47 "There are different technologies to convert gas to electricity including gas turbine in Brayton cycle, Organic Rankine cycle, Stirling cycle engine [3-4]". 

Gas engines are usually the most popular way of generating electricity from landfill methane gas. The ones mentioned are rather specialised.

There are examples of landfill methane gas recovery and energy use around the world. It would be interesting to know what the challenges are in those global cases and how your approach can contribute to them.

Author Response

(The authors gave the same response as above.)

Reviewer 3 Report

This is a very interesting topic, the publication is well put together and extremely valuable. The topic makes strong practice-oriented and valuable suggestions.

Author Response

(The authors gave the same response as above.)

Round 2

Reviewer 2 Report

I have checked the "Author response". I think it has been properly fixed.